# Improvement of *Pacific* White Shrimp (*Litopenaeus vannamei*) Drying Characteristics and Quality Attributes by a Combination of Salting Pretreatment and Microwave

**DOI:** 10.3390/foods11142066

**Published:** 2022-07-12

**Authors:** Yawen Lin, Yue Gao, Aiqing Li, Lei Wang, Ziping Ai, Hongwei Xiao, Jianrong Li, Xuepeng Li

**Affiliations:** 1National R&D Branch Center of Surimi and Surimi Products Processing, College of Food Science and Engineering, Bohai University, Jinzhou 121013, China; linyawen2020@163.com (Y.L.); gy1362070342@163.com (Y.G.); liaiqing3483@163.com (A.L.); lijr6491@163.com (J.L.); 2National & Local Joint Engineering Research Center of Storage Processing and Safety Control Technology for Fresh Agricultural and Aquatic Products, College of Food Science and Engineering, Bohai University, Jinzhou 121013, China; 3School of Liquor and Food Engineering, Guizhou University, Guiyang 550025, China; wanglei6166@126.com; 4College of Engineering, China Agricultural University, Beijing 100083, China; aiziping@cau.edu.cn (Z.A.); xhwcaugxy@163.com (H.X.); 5Collaborative Innovation Center of Seafood Deep Processing, Dalian Polytechnic University, Dalian 116034, China

**Keywords:** salting pretreatment, microwave drying, water distribution, protein secondary structure, astaxanthin

## Abstract

This study investigated the effects of salting pretreatment and microwave (MW) power on drying characteristics, water distribution and quality attributes of *Pacific* white shrimp *(Litopenaeus vannamei)*. With increasing salt concentration (0–8%, *w*/*v*) and MW power (300–900 W), the drying time of shrimp was shortened by 15.15–28.57%, compared with the untreated samples. Regarding the quality of dried shrimp, increasing the salt concentration and MW power increased the hardness (from 13,073.6 to 24,556.5 g), while the springiness, color parameters and astaxanthin content showed an initial decrease but a later increase trend. Low field nuclear magnetic resonance (LF-NMR) demonstrated that the T_2_ curve of the pretreated samples moved toward the negative *x*-axis and the immobilized water content decreased with increasing salt concentration. E-nose showed that volatile components were different and could be obviously distinguished at different salt concentrations and MW powers. Raman spectroscopy illustrated that the protein secondary structure of dried shrimp was altered by salting pretreatment and drying conditions, and the lowest conversion degree of α-helix to β-sheet of dried shrimp was obtained at the salt concentration of 4% (*w*/*v*) and MW power of 500 W. By comprehensively considering the drying time and quality attributes, the combination of 4% (*w*/*v*) salt and 500 W MW power was concluded as the best drying conditions for shrimp using a microwave. The results could provide an innovative combination of salt pretreatment and MW drying with suitable processing conditions for producing high-quality dried shrimp.

## 1. Introduction

Shrimp plays an important role in the aquaculture industry in China, as well as across the world. According to the statistics of the Food and Agriculture Organization (FAO), the global white shrimp production reached 4.5 million tons in 2019, of which China (1800 thousand tons), India (720 thousand tons), Indonesia (700 thousand tons), Ecuador (680 thousand tons), and Vietnam (570 thousand tons) ranked as the top five countries worldwide [1]. *Litopenaeus vannamei* (*Pacific* white shrimp) and the other two farmed shrimp, including *Penaeus chinensis* and *Penaeus monodon,* are the top three most popular species in the world. *Pacific* white shrimp has been widely cultivated and consumed in China due to its disease-resistant, high-yielding characteristics and rich nutrients including well-balanced amino acids, unsaturated fatty acids, and antioxidants, such as astaxanthin and vitamin B_12_ [2,3].

However, similar to other fresh shrimp, *Pacific* white shrimp shows high moisture content (~76.05%) and is susceptible to deterioration. Therefore, fresh shrimps should be stored under frozen conditions or/and processed immediately after harvesting to extend the shelf life [4]. Although frozen storage can preserve the freshness and nutritional value of fresh shrimp, it is expensive and difficult to maintain the quality for a long time. Drying is the most commonly used preservation method to prevent quality loss and product waste of fresh shrimp by preventing microbial growth and enzymatic activities. It can not only maximize the shelf life of aquatic products, but also can reduce the cost of packing, storage, and transportation, as well as improve the commercial value of dried products [5]. Hence, various drying methods have been developed to dry aquatic products, such as hot air drying (HAD), cold air drying (CAD) and vacuum freeze drying (VFD) [6,7,8]. HAD and CAD are normally applied in the dehydrated aquatic products industry. However, they often result in product quality deterioration, such as the loss of nutrients and bad taste due to the long processing time [9]. Various studies showed that HAD can also cause an unacceptable color in products by inducing fat oxidation, Maillard browning, surface hardening and cracking, excessive shrinkage and low rehydration [10,11,12]. Compared with HAD or CAD, VFD can preserve nutritional quality and original color of foods due to being processed under a vacuum and low temperature, but the high energy consumption, investment and operation cost as well as the long drying time limit its industrial application [13]. As a novel drying method, microwave (MW) drying converts electromagnetic energy into heat energy to heat materials volumetrically [14]. Due to its fast drying rate and being easy to operate, it has been widely used in drying and processing of cereals, fruits, vegetables and aquatic products [15,16]. Kipcak and İsmail [17] studied the effects of MW drying power on the drying kinetics of fish and observed that drying time decreased as MW power increased, and the MW power level was the main factor affecting the color change in the treated material. Pankyamma et al. [18] explored the effects of microwave vacuum drying (MVD), HAD and sun drying (SD) on the basic composition, texture, and color of squid silk, and found that lightness of MVD samples was higher than that of SD and HAD samples, while hardness was higher for SD samples. Wan et al. [19] used HAD and MVD to dry salted grass carp fillets, and found that the MVD had a higher crude fat content (dry basis) because the absence of air may prohibit oxidative reactions and greatly reduced drying time, due to increased effective moisture diffusivity. These findings demonstrate the feasibility of MVD in seafood drying. Nevertheless, one of its noticeable shortcomings was the unavoidable non-uniformity of the electromagnetic field, which affected the quality of food products, and hence limited the application of MVD in the food industry.

Pretreatment prior to drying is often utilized to accelerate the drying process, enhance product quality and reduce energy consumption. Several pretreatments have been employed to improve quality, reduce nutrient loss and extend the shelf life of dehydrated products, such as blanching [20], chemical treatment [21], ultrasound [22], high pressure [7], osmotic dehydration (OD) [23]. Li et al. [24] found that ultrasound-assisted osmotic pretreatment significantly improved the quality of dried tilapia fillets with higher rehydration rates and Ca^2+^ adenosine triphosphate synthase (Ca^2+^-ATPase) activity after heat pump drying. Ling et al. [7] monitored the moisture migration in shrimps during high pressure processing-vacuum-freeze drying (HPP-VFD) processes by using low-field nuclear magnetic resonance and magnetic resonance image and compared with direct HAD and VFD. The results showed that HPP pretreatment could shorten the relaxation time and improve drying efficiency. Aquatics infiltration treatment using the semi-permeable nature of the cell membrane to transfer the water to the solution could achieve the purpose of removing part of the water [25]. Salt as a solution can also accelerate the drying rate of subsequent drying and improve the organizational structure, flavor quality, processing efficiency and physical and chemical characteristics of food to a certain extent. Niamnuy et al. [26] studied the concentration of salt solution (2%, 3%, 4%) on the drying kinetics and quality attributes of shrimps by a jet-spouted bed dryer, and found that higher concentrations of salt solution can lead to higher drying rate of shrimps and higher toughness and shrinkage.

However, for whole shrimp drying, the hard shell would prevent the internal moisture migration of shrimp body to the outside, and shrimp body thickness is inconsistent and there is an uneven distribution of moisture content, exacerbating the unevenness of microwave heating. Salting pretreatment could increase the salt content of shrimp, improve the drying rate and reduce drying time. To the best of our knowledge, no reports have been found exploring the effects of salting treatment prior to MW drying on the drying characteristics, water distribution and quality attributes of *Pacific* white shrimps. This study evaluates the effects of salt pretreatment (0, 2%, 4%, 6%, 8%) combined with MW powers (300, 500, 700, and 900 W) on the quality and the drying rate of shrimp. Several parameters, including texture properties, color, astaxanthin content, and volatile components, etc. of dried *Pacific* white shrimps, under different conditions were measured. Moreover, moisture distribution, microstructure and protein secondary structure were analyzed to provide a theoretical basis to select the suitable drying process parameters of shrimp.

## 2. Materials and Methods

### 2.1. Sample Preparation

Fresh *Pacific* white shrimps were purchased from a local seafood market in Jinzhou, China, and stored in a refrigerator at −18 °C for no more than 30 days. Prior to the experiment, the samples were thawed at 4 °C for 4 h for the subsequent experiments. The initial moisture content of the samples was 76.05% ± 1.05% in wet basis (w.b.). Shrimps (15–18 g/each) were immersed in brine with a salt concentration of 0, 2%, 4%, 6% or 8% (*w*/*v*) in a 1:6 ratio for 8 h at 4 °C [27], and mixed at 2 h intervals. Then, the pretreated samples were dried in a MW oven with power of 300, 500, 700 and 900 W; the drying process was terminated when the moisture content was below 16% (wet basis). When the MW power is 300, 500, 700 and 900 W, the drying time is 28–33 min, 20–30 min, 9–10 min and 6.5–8 min, respectively. At this time, the water activity (a_w_) of dried shrimp was 0.63 ± 0.02, which can be stored for long time with a good appearance [26].

### 2.2. Drying Properties

During the drying process, the mass of shrimps was weighted using an electric balance (JCS-31002C, ZOGG, Shanghai, China), and the moisture content (g/g d.b.) was calculated using Equation (1).
(1)$$M_{t}=\left(m_{t}-m_{d}\right)/m_{d\;}$$
where $M_{t}$ is the moisture content (g/g d.b.) at the time *t*, $m_{t}$ is the mass (g) of shrimp at time *t*, and $m_{d}$ is the final dry mass (g) after drying treatment until a constant weight.

The moisture ratio (*MR*) of the shrimps was calculated with Equation (2) [27].
(2)$$MR=\frac{M_{t}}{M_{0}}\;$$
where *M*_0_ is the initial moisture content (g/g, d.b.) of fresh shrimp.

### 2.3. Water State and Distribution by Low-Field Nuclear Magnetic Resonance (LF-NMR)

Dried shrimps were cut into 1 cm × 1 cm × 2 cm cubes. The samples were placed in NMR tubes with a diameter of 1.5 cm and a height of 20 cm and then placed in a LF-NMR analyzer (NMI 2012, Shanghai Niumai Analytical Instrument Co., Shanghai, China) to collect Carr–Purcell–Meiboom–Gill sequence (CPMG) decay signals according to the method described by Wang et al. [28], with slight modifications. The transverse relaxation measurements of shrimp were performed on a MesoQMR23-060H NMR analyzer (Suzhou Niumag Analytical Instrument Corporation, Suzhou, China) equipped with a vertical 0.5 T permanent magnet at 32 °C, according to a resonance frequency for protons of 22.0 MHz. The lengths of 90 and 180 pulse were 18.0 and 36.0 ms, respectively. The repetition time between two scans was 2500 ms, and the data from 6000 echoes were acquired as 4 scan repetitions. Every measurement was performed in triplicate. 

### 2.4. Determination Salt Content 

The salt content of the salted shrimp was measured according to the aquatic industry standard of the People’s Republic of China “Determination of salt In aquatic products” in the direct titration method.

We absorbed the water on the surface of the sample with filter paper, removed the salt impurities from the surface (until invisible to the naked eye), cut these into small pieces below 5 mm × 5 mm and mixed well. We weighed 20 g of the sample (weighed to 0.01 g) in a 250 mL beaker, added 150 mL of water, heated and boiled it, allowed it to cool naturally, transferred the liquid into a 500 mL volumetric flask, then rinsed the residue with 50 mL of water three times, combined the washes in the same volumetric flask, cooled and diluted it to the scale with water and set it aside. We took a certain amount of sample solution in a 250 mL triangular flask and added water to about 100 mL, added 2~3 drops of thymol blue indicator, titrate with 0.1 mol/L sodium hydroxide until just light blue (pH between 6.5~10.5), added 0.5 mL of 10% potassium chromate indicator, titrate with 0.1 mol/L silver nitrate standard solution until it showed brick red as the end point, and we performed this with the blank control at the same time.

The salt content in the sample was calculated according to Equation (3). The calculation results are retained to the second decimal place.
(3)$$X\left(\mathrm{in}\;\mathrm{NaCl},\%\right)=\frac{\left(V-V_{0}\right)\times c\times 0.05845}{m\times \frac{V_{1}}{V_{2}}}\times 100$$
where *X* is the salt content of the sample (in NaCl, %); *V* is the volume of silver nitrate standard solution used for titration of the sample, mL; *V*_0_ is the volume of silver nitrate used for titration blank, mL; *V*_1_ is the volume of filtrate removed used for titrate, mL; *V*_2_ is the total volume of the sample after treatment, mL; *c* is the concentration of silver nitrate standard solution, mol/L; *m* is the weigh the mass of the sample, g; 0.05845 is the mass of sodium chloride equivalent to 1 mL, 1 mol/L silver nitrate standard solution, g.

### 2.5. Determination of Rehydration Ratio

The rehydration ratio (RR) of dried shrimp was measured according to the experimental operation by Wei et al. [29]. Briefly, approximately three dried shrimps were rehydrated by soaking in 200 mL deionized water at 100 °C for 10 min, the samples were then withdrawn, drained, gently blotted with paper towels to eliminate the surface water and weighed. The rehydration ratio was calculated using Equation (4).
(4)$$R_{f}=\frac{m_{f}}{m_{g}}$$
where *m*_f_ is the weight (g) of the samples after rehydration, and *m*_g_ is the weight of the samples before rehydration.

### 2.6. Texture Measurement

The texture profile analysis of the dried shrimp was carried out using a TA-XT Plus texture analyzer (TA-XT plus, Stable Micro Systems, Godalming, UK). The analyzer was performed with a compression mode and the operating conditions were as follows: interval between the two compressions was 5 s; compression degree was 30%; a flat-bottomed cylindrical probe (50 mm in diameter) was selected; the pre-test speed, test speed and post-test speed of the probe was set at 1 mm/s.

### 2.7. Color Measurement

The color parameters of the dried shrimp were determined using a colorimeter (CR-400, Konica Minolta, Tokyo, Japan). CIE Lab coordinates were reported as lightness (*L**), redness (*a**), and yellowness (*b**). Total color change (Δ*E*) was calculated according to following Equation (5) [30,31]:(5)$$\Delta E=\sqrt{(L^{*}-L_{0}^{*}{)}^{2}+(a^{*}-a_{0}^{*}{)}^{2}+(b^{*}-b_{0}^{*}{)}^{2}}$$
where $L_{0}^{*}$, $a_{0}^{*}$, and $b_{0}^{*}$ represented the color parameters of fresh shrimps.

### 2.8. Determination of Volatile Vomponents by E-Nose

A portable e-nose system (PEN3, Win Muster Airsense Analytics Inc., Schwerin, Germany) was used to acquire the aromatic information and volatile compound data. For the measurements, about 2 g of the sample was placed in a 50 mL centrifuge tube and sealed the top with plastic wrap. After that, the headspace of the sample was equilibrated for 30 min at a constant temperature to minimize sensor drift caused by environmental changes. Five replicates were completed for each group. Principal component analysis (PCA) was used to study the statistically significant difference in the mean of each sensor (*p* < 0.05).

### 2.9. Determination of Astaxanthin Content

The procedure for astaxanthin content measurements was carried out based on the method by Tolasa et al. [32] with slight modifications. Samples (about 10 g) were extracted three times with 40 mL of acetone solution using a homogenizer (JHG-Q60-P100, Ronghe, Shanghai, China) for 2 min. Samples were kept cool by immersing in crushed ice during homogenization to avoid over-heating. After extraction, the samples were centrifuged at 4000× *g* for 5 min at 4 °C. To separate the water insoluble compounds, the acetone extracts from the samples were transferred to a 250 mL separating funnel with 40 mL of petroleum ether. Then, 100 mL of distilled water containing 0.5% *w*/*v* sodium chloride were added to the mixture. After continuous manual shaking, the phase separation was achieved, so the upper layer was removed and transferred into a 50 mL volumetric flask. The absorption spectrum of the water insoluble compounds was recorded at 472 nm using a photodiode array spectrophotometer (Shimadzu Scientific Instruments, UV2550, Kyoto, Japan). A standard curve of astaxanthin (y = 0.2037x + 0.0152, R^2^ = 1) was prepared according to Tolasa et al. [32]. The results were expressed as per gram db (mg/g db).

### 2.10. Determination of Protein Secondary Structure by Raman Spectroscopy

Protein secondary structure was analyzed using a confocal Raman spectrometer (Lab RAM HR Evolution, HORIBA, Ltd., Shanghai, China) with a 532 nm argon ion laser and lines 600 grating. The samples were cut into slices with thickness of 2 mm, and then put on the glass sheet. Raman spectra were scanned in the range of 400–3600 cm^−1^ with the resolution of 1 cm^−1^. The acquisition time was 60 s. Percentages of protein secondary structures were calculated using the method reported by Alix et al. [33].

### 2.11. Determination of Microstructure

After drying, the heads and shells of the shrimps were removed; the 3rd abdominal segment of the dried shrimp was cut off and cut into thin slices. The samples were soaked in 2.5% glutaraldehyde solution for 5 h, then rinsed three times with sodium phosphate buffer pH 6.8, 0.1 mol/L, for 15 min each, followed by one rinse with 50%, 70%, 80%, and 90% ethanol solution for 10 min each, and finally three rinses with 100% ethanol for 10 min each. After the samples were rinsed, they were micro frozen in clean and dry Petri dishes. The dried samples were observed by scanning electron microscopy (JSM-IT200, JEOL, Ltd., Beijing, China) [18]. 

### 2.12. Statistical Analysis

Data were subjected to one-way analysis of variance (ANOVA). The mean multiple comparison tests were conducted by Duncan’s multiple range tests with a significant level of 5%. All the experimental data were represented by mean ± standard deviation. Statistical analysis was performed by IBM SPSS 22.0 (SPSS Inc., Chicago, IL, USA). PCA analysis data were performed using the Winmuster software that comes with the electronic nose (e-nose) equipment.

## 3. Results and Discussion

### 3.1. Drying Kinetics

Changes in moisture content versus drying time at different MW powers and salt concentrations are shown in Figure 1. When the microwave power is 300, 500, 700 or 900 W, the drying time is estimated to be 28–33, 20–30, 9–10 or 6.5–8 min, respectively. The results showed that MW power and salt treatment significantly reduced the drying time of shrimp. When the salt concentration increased from 0 to 8%, the drying time of shrimp was shortened by 10–33% at different levels of MW power. When a higher level of MW power was applied, the MW energy absorbed by the whole shrimp per unit volume was greater, which led to the faster oscillation rate of water in the whole shrimp and faster evaporation rate of water molecules [34]. Wan et al. also found the same phenomenon in the experiment of vacuum MW drying of salted grass carp slices [19]. The potential explanation is that the MW could cause high-speed movement of water molecules and ions due to the increasing salt concentration inside the shrimp, which finally increased the drying rate of shrimps.

As shown in Figure 1, when the MW power increased from 300 to 900 W, the drying time of shrimp was shortened by 75% with the increasing salt concentration. Salting pretreatment has a certain influence on the moisture content during the drying process of shrimp. The moisture content of shrimp dipped above 4% salt concentrations decreased quickly because of the NaCl osmotic dehydration effect on the shrimp. Salinization can decompose part of the salt-soluble protein and fat in the prawn body, resulting in the change in water locked inside the prawn body. During the drying process of the shrimp, salting pretreatment can improve the drying rate. In the drying experiment of sardines, Bellagha et al. [35] found that the migration rate of water molecules from the inside to the outside was related to the increase in salt concentration. The water migration rate was faster with a higher salt concentration. It shows that the MW power is the main factor affecting the drying rate, and salting pretreatment can effectively improve the drying rate of shrimp. The appropriate combination of MW power and salt concentration can be chosen according to the actual requirements.

### 3.2. Moisture State and Distribution

The transverse relaxation time T_2_ and signal amplitude A_2_ are important outcome parameters that reflect the state of water in the material. Three distinct water populations centered at 1–10 ms (T_21_), 10–100 ms (T_22_), and 100–1000 ms (T_23_) were observed in all samples treated by salting and drying (Figure 2a–c). T_21_ represents the bound water associated with or trapped within highly organized structures. T_22_ stands for the immobilized water entrapped within the myofibrillar structure. T_23_ is ascribed to free water in the myofibril lattice [36]. As shown in Figure 2, after the shrimp were pretreated with salting, the free water relaxation time shifted to the right and the relaxation time of less mobile water shifted to the left with the increasing salt concentration, indicating that salting promoted the dissipation of free water and the conversion to less mobile water. It clearly showed that with the increase in salt concentrations and MW powers, the peak area gradually decreased and the T_2_ distribution curves gradually moved towards the left *x*-axis, which meant the combination degree of water and non-water components became more tight. The immobilized water was the main water of fresh shrimp and bound water was the main water in the dried shrimp samples, which showed that the mobility of immobilized water reduced gradually after drying. The possible reason was ascribed to the gradual shrinkage of the myofibrillar structure in shrimps during drying [36]. When the MW power increased from 300 to 900 W, the lateral relaxation time (T_21_) decreased from 4.04 to 2.66 ms. The peak area for T_21_ (A_21_ in Table 1) also decreased upon increasing MW power. It was concluded that the increased MW power released more MW energy, which can be converted to thermal energy rapidly, resulting in a larger amount of water loss and shorter T_2_ relaxation time. When the salt concentration increased from 0% to 8%, the lateral relaxation time (T_21_) decreased from 4.04 to 2.00 ms. This was attributed to the denaturation of proteins during the soaking in brine, which in turn led to the lower water-holding capacity of the shrimp samples [37].

The signal amplitude (A_2_) has been proved to be proportional to the amount of water in the food sample. It was noted that three water fractions with different molecular environments were observed in dried shrimps, and the corresponding information of the peak area per mass expressed as A_21_, A_22_, and A_23_ are summarized in Table 1. There was significant difference (*p* < 0.05) among the bound water content of the dried shrimps with different MW powers and salt concentrations. It was noted that the area of the bound water (A_21_) after being salted at 700 and 900 W was lower than that at 300 and 500 W, and the area of the bound water decreased after being salted at 6% and 8%, and the reason was ascribed to the loss of bound water.

### 3.3. Salt Content of Shrimps Pretreated with Different Salt Concentrations

The salt content is an important index for evaluating salted products, and the amount of salt content is closely related to the sensory quality, storage time and safety of the products [38]. As shown in Table 2, after 8 h of salt soaking treatment, the salt content of shrimp increased significantly (*p* < 0.05) from 0.54% to 1.83% when the salt concentration increased from 0 to 8%. This is due to the low osmotic rate of the low concentration of brine, and the higher the concentration of brine, the greater the osmotic rate. The lower the concentration, the shorter the time for salt osmosis to reach equilibrium. 

### 3.4. Rehydration Ratio of Dried Shrimps Treated by Bifferent Salt Concentrations and MW Powers

As shown in Table 2, there was no significant effect in the salt concentration on the rehydration rate of dried shrimp samples. MW power had a significant (*p* < 0.05) effect on the rehydration rate of dried shrimp samples, and the maximum rehydration ratio was obtained when the MW power was 700 W, which may be attributed to the fact that under low MW power, longer drying time, in the late stages of drying, where the material internal free moisture is basically dissipated, microwave heating on the solid tissue of the material for a longer period of time, there may be internal overheating, meaning that localized coking has occurred, which is not visible in the appearance of the situation, thus limiting its ability to rehydrate, resulting in a lower rehydration rate [14]. 

### 3.5. Color

Color is an important indicator for quality grade and dried shrimps with bright red color can be accepted more easily by customers. The color changes in dried shrimps under different salt concentrations and MW powers were shown in Figure 3. The *L** value increased gradually; the *a** and *b** value first increased and then decreased with the extension of the salt concentration. *L** reached the highest value in 8% salt concentration; both *a** and *b** almost reached the highest values in 4% salt concentration with different MW powers. During the drying process, due to the irreversible denaturation of protein and the contact with oxygen, a Maillard reaction occurred, resulting in a dark yellow surface, while the increase in salt concentration inhibited the color browning of dried shrimps [39]. At a fixed salt concentration, the *a** values first increased and then decreased, and the *b** values decreased with increasing MW powers. The color difference (∆*E*) was larger at a salt concentration of 2–6% of dried shrimp, while the MW power had no significant effect on the color difference of shrimp. This indicated that a shorter drying time can help avoid oxidation reactions during MW drying. Indeed, ∆*E* showed a positive correlation (*p* < 0.05) with DT, as shown in Figure 4. Meanwhile, astaxanthin content showed a positive correlation (*p* < 0.05) with the *a** value. Suitable salt concentration as well as MW power can improve shrimp color through protein denaturation or Maillard reaction, resulting in higher astaxanthin content [40]. It can be further concluded that MW drying in reasonable MW intensity, such as 500 W, can help retain a better color of dried shrimps. 

### 3.6. Volatile Components 

E-nose technology is a non-destructive technology with simple, fast and accurate operation. It uses the response curve of gas sensor arrays to identify volatile odors of samples, and has advantages of high sensitivity and good reproducibility [41]. Taking the MW power of 300 W as an example, the effects of different salt concentrations on volatile components of dried shrimps were studied and shown in Figure 5A. The accumulative variance contribution ratio of the two main components (PC_1_ + PC_2_) reached more than 90%, indicating that the information of these two components was sufficient to show the effect of salt concentration on the volatile components. There were no overlapping regions in the odor response values of the five salt concentrations, indicating that the volatile flavor components of dried shrimps were different under different salt concentration pretreatment and could be distinguished by electronic nose with a good degree of distinction. 

Taking the salt concentration of 4% as an example, the effects of different MW powers on the volatile components of dried shrimps were studied and shown in Figure 5B. The contribution rates of the first and second principal components in this study were 82.24% and 11.16%, respectively, and the total contribution rate reached 93.401%. The results showed that the electronic nose could well distinguish the volatile components of dried shrimps with different MW powers. The volatile flavor components of shrimps were different under different MW powers and could be well distinguished by electronic nose.

### 3.7. Protein Secondary Structure Analysis

Raman spectra can provide the micro-environmental chemical information of protein side chains and polypeptide chain configurations, which is mainly derived from the stretching vibration of the C–N bond and C=O bond in protein molecules [42]. The Raman spectra at 400–4000 cm^−1^ is shown in Figure 6A and the ratios of the second structure of the protein are shown in Figure 6B. The change in the amide I (1600–1700 cm^−1^) was closely related to the change in the content of various secondary structures of the protein. Protein secondary structure referred to the structure formed by the skeleton of some peptide segments in the peptide chain; it mainly involved α-helix (1645–1657 cm^−1^), β-sheet (1665–1680 cm^−1^), β-turn (near 1680 cm^−1^) and random coil (1660–1665 cm^−1^). The α-helix and β-sheet characterized the regularity of protein molecules, and the β-turn and random coil usually reflected a looser structure. With the increase in MW power, β-turn and random coil of dried shrimps remained unchanged, α-helix first increased and then decreased. In the process of protein heating, the hydrogen bonds ruptured and the polypeptides gradually unwound and became a loose random structure [43], resulting in the content of α helix being decreased and the content of β-sheet being increased. With the increase in salt concentration, the second structure of the protein showed the same pattern. There was less α-helix transforming into β-sheet when the salt concentration was 4% and the MW power was 500 W, suggesting that the samples showed a more stable protein structure under this condition. 

### 3.8. Astaxanthin Analysis

Figure 7 showed the astaxanthin content of dried shrimps under different drying and pretreatment conditions. It can be observed that astaxanthin content in dried shrimps was affected significantly by the salt concentration and MW power (*p* < 0.05). Astaxanthin content increased at first then decreased with increasing salt concentrations and MW powers. It reached the highest value of 25.95 μg/g (dry weight) and the lowest value of 19.26 μg/g (dry weight) when the salt concentration was 4% and 0%, respectively. This may be due to the changes in structural conformations induced by salinization and the protein lost its binding affinity to small molecules including astaxanthin, thus increasing the content of free astaxanthin. However, at the same time, astaxanthin oxidized by thermal oxidation [44] and higher salt concentrations led to an increased degree of protein denaturation, and when the part of the protein bound to astaxanthin completely denatured, the degradation rate of astaxanthin would be greater than the formation rate, resulting in a decrease in the astaxanthin content at high salt concentrations (≥6%). This phenomenon can also be confirmed by the changes in protein secondary structure in Figure 5, where the degree of protein denaturation increased when the salt concentration increased from 4% to 8%. The astaxanthin content at different MW powers decreased from 25.95 μg/g (dry weight) at 500 W to 22.93 μg/g (dry weight) at 900 W. More astaxanthin degradation occurred due to the long drying time when the MW power was 300 W. With the increasing MW intensities from 500 to 900 W, the levels of total astaxanthin decreased. Yang et al. [45] monitored the change in total astaxanthin in *Pacific* white shrimp under different microwaving conditions, and found that the content of total astaxanthin decreased as the treatment time increased from 0 to 20 min. The total astaxanthin content decreased by 10.69% and 19.24% (200 W), 14.88% and 50.06% (400 W), and 47.40% and 63.59% (600 W) after 3 and 10 min, respectively. It was shown that the increase in microwave power and microwave time accelerated astaxanthin degradation. In this paper, the MW power of 500 W and salt concentration of 4% were suitable for preserving the higher astaxanthin content of dried shrimps.

### 3.9. Microstructure of Dried Shrimps under Different Salt Concentrations and MW Powers

The SEM images in Figure 8A visually showed the changes in the microstructure of shrimp tissues subjected to the salt concentrations 0%, 2%, 4%, 6% and 8% at the MW power 500 W. As can be observed from the Figure 8A, for the control sample (at 0% salt concentration), shrimp muscle fiber bundles were arranged parallel to each other with large gaps between the muscle fiber bundles and intact muscle bundle membrane structure. However, with increasing salt concentration (from 2% to 8%), the muscle fiber bundle was destroyed with marked contractions and a large number of gaps. The reason for this may be that the rate of salt penetration into the muscle fiber cells during salting was less than the rate of water and other soluble component exudation from the muscle fiber cells, thus making the muscle fibers contract and the cross-sectional fibers showed different shapes and sizes [46]. The SEM images in Figure 8B visually showed the changes in the microstructure of shrimp tissues subjected to MW power 300 W, 500 W, 700 W and 900 W at a fixed salt concentration of 4%. As can be observed from the Figure 8B, with the increase in microwave power, the degree of damage to the muscle fiber increased, resulting in increased gaps between the muscle fiber bundles, which further explained the higher drying rate of shrimp under high MW power conditions (Section 3.1).

### 3.10. Texture

The texture properties of the samples with different MW powers and salt concentrations are presented in Table 3. It can be concluded that both salt concentration and MW power had a significant effect on the texture properties (*p* < 0.05). With the increase in salt concentration, the hardness and chewiness of the dried shrimp gradually increased, while the springiness first decreased and then increased. This was because the salt permeated into the flesh of the shrimp and made the structure of the shrimp muscle protein more compact. In addition, protein denaturation, water loss and muscle fiber aggregation also occurred during the drying process [47]. With increased MW power, the hardness of the shrimp increased and this may be related to the fact that the application of MW could contribute to some mild damage in the protein structure, causing a degree of hardening [48]. Moreover, the correlation analysis (Figure 4) showed that drying time was significantly negatively correlated (*p* < 0.05) with hardness and chewiness, indicating that different drying times could affect the textural properties under different salt concentrations and MW powers.

## 4. Conclusions

Based on the above research results, we can conclude that salting pretreatment can effectively improve the MW drying rate and quality of Pacific white shrimps. With the increase in salt concentration and MW power, the drying rate of the shrimp improved. The *a** values increased firstly and then decreased and the *b** values decreased with increasing salt concentrations. The hardness of the dried shrimps increased with increasing salt concentrations and MW powers, while the springiness firstly decreased and then increased with increasing salt concentrations. The highest astaxanthin content can be preserved at the salt concentration of 4% and the MW power of 500 W. β-turn and irregular coiling presented no significant difference at different MW powers or salt concentrations, and the α-helix firstly decreased and then increased with increasing salt concentrations. Different drying conditions can lead to different volatile components. The processing condition of salt concentration of 4% and MW power of 500 W was considered as the best processing parameter due to better color appearance, texture, as well as less protein denaturation. This study demonstrated that salting pretreatment combined with MW drying is a promising method for *Pacific* white shrimp drying.

This study provided the basis for the feasibility of salt pretreatment and microwave drying of *Pacific* white shrimp with low costs and simple preparation. In future, some antioxidants could be added to improve the color appearance and quality of dried shrimps.

## Figures and Tables

**Figure 1 foods-11-02066-f001:**
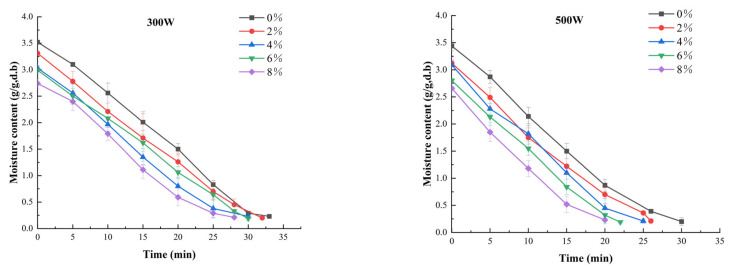
Effect of different salt concentrations and microwave power on drying rate of white shrimp.

**Figure 2 foods-11-02066-f002:**
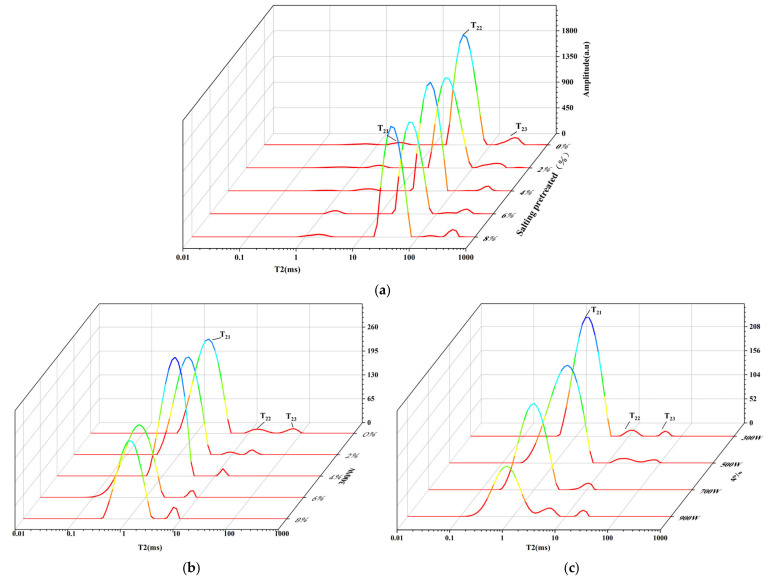
T_2_ relaxation time distribution curves of dried shrimps by different drying treatments: fresh shrimps with different salt concentrations (**a**), dried shrimps with different salt concentrations (**b**), dried shrimps with different MW powers (**c**).

**Figure 3 foods-11-02066-f003:**
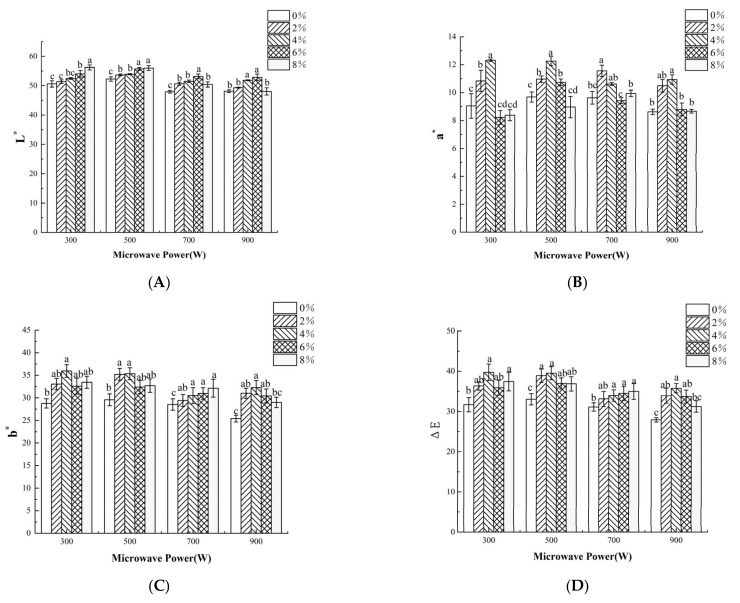
Effects of salt concentration and MW power on color parameters of dried shrimps by MW drying. (**A**) *L**; (**B**) *a**; (**C**) *b**; (**D**) Δ*E*. Different letter indicates statistically significant difference at *p* < 0.05 according to the Duncan test.

**Figure 4 foods-11-02066-f004:**
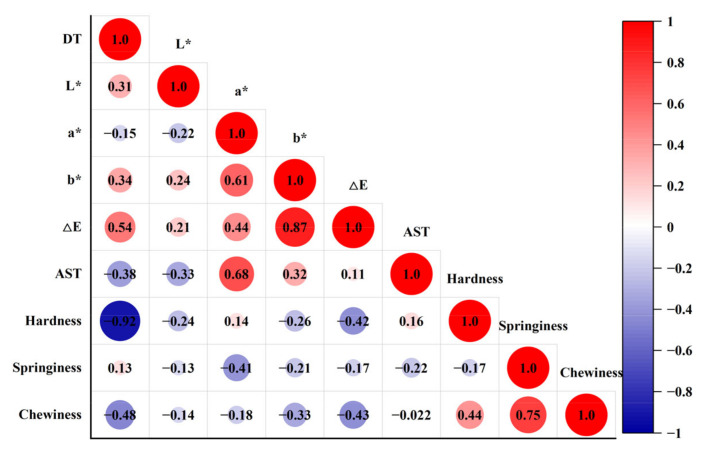
Correlation matrixes between the determination parameters. DT: drying time; AST: astaxanthin.

**Figure 5 foods-11-02066-f005:**
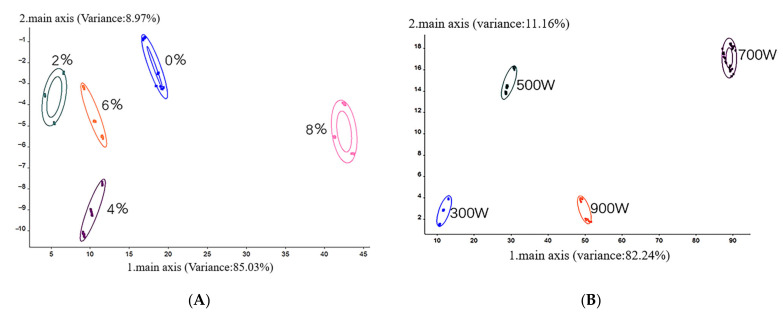
PCA diagram of volatile components in dried samples with different salt concentrations (**A**) and MW powers (**B**).

**Figure 6 foods-11-02066-f006:**
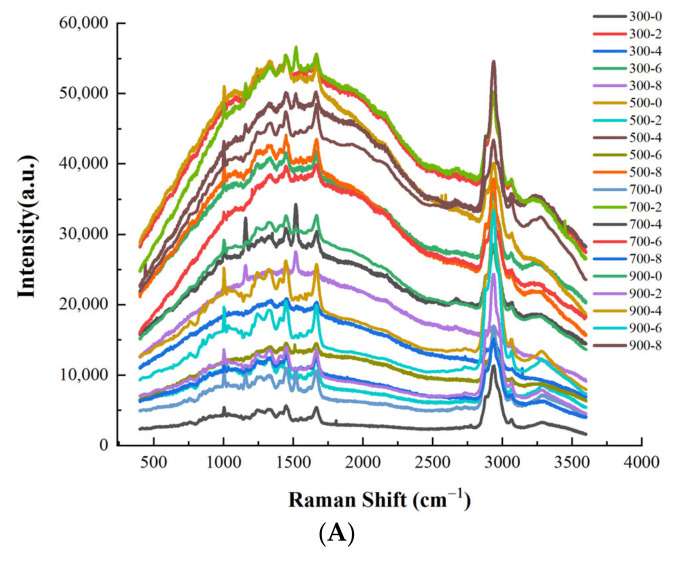
Raman spectra (**A**) and different proportions of secondary structures of proteins under different heating treatments of dried shrimps (**B**) (a.u.: arbitrary units). Different letter indicates statistically significant difference at *p* < 0.05 according to the Duncan test.

**Figure 7 foods-11-02066-f007:**
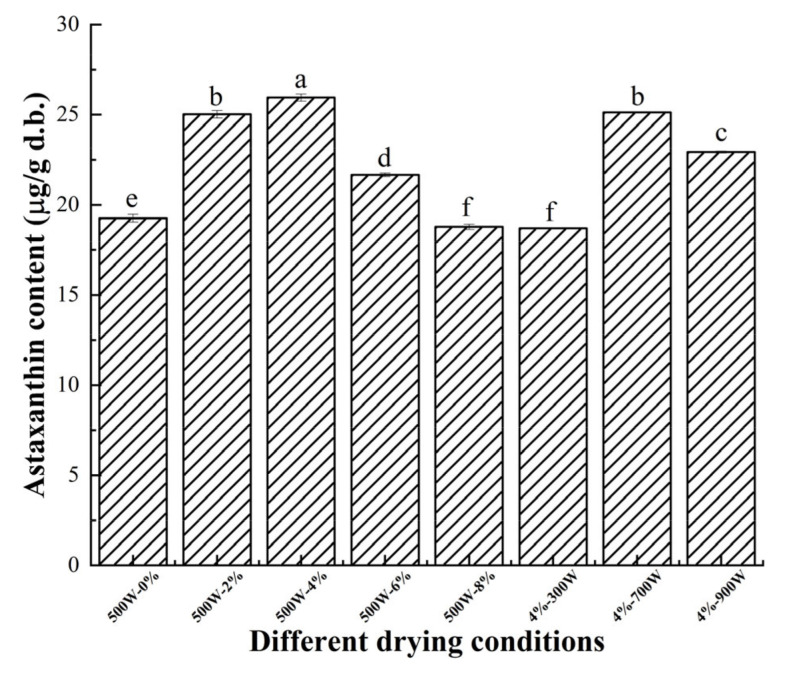
Changes in total astaxanthin content (μg/g) (dry weight) of shrimps after MW drying under different salt concentrations and MW powers. Different letter indicates statistically significant difference at *p* < 0.05 according to the Duncan test.

**Figure 8 foods-11-02066-f008:**
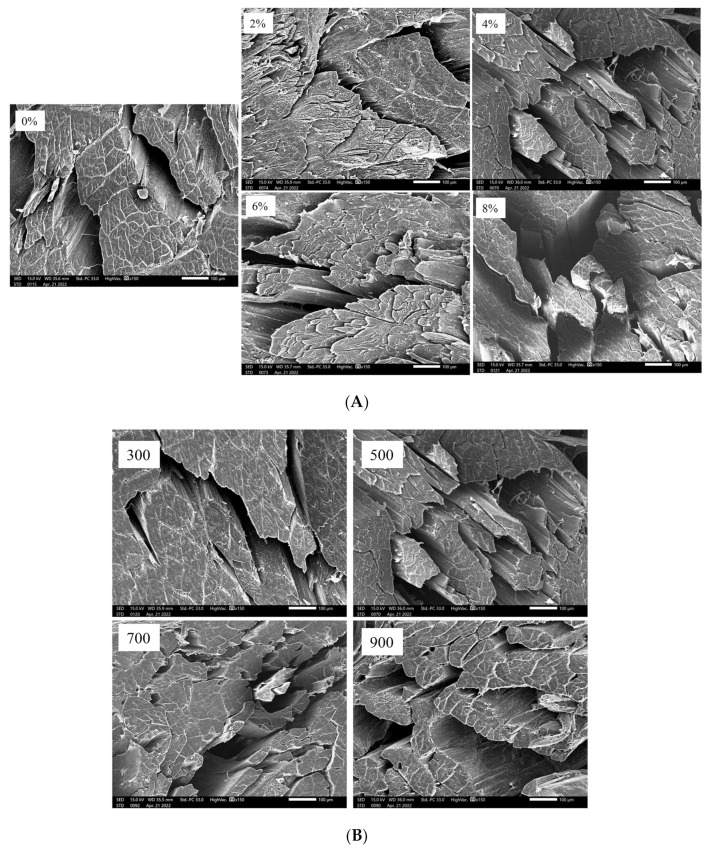
Effect of different salt concentrations on the microstructure of dried shrimps (500×) at a fixed MW of 500 W (**A**); the effect of different microwave powers on the microstructure of dried shrimp (500×) at a fixed salt concentration of 4% (**B**).

**Table 1 foods-11-02066-t001:** Peak area relaxation populations of dried shrimps with different salt concentrations and MW powers.

Treatment Conditions		A_21_/%	A_22_/%	A_23_/%
Salt pretreatment	0%	2.31 ± 0.44 ^ab^	92.69 ± 0.64 ^b^	5 ± 0.55 ^a^
	2%	2.73 ± 0.27 ^a^	92.8 ± 0.19 ^b^	4.47 ± 0.36 ^a^
	4%	2.4 ± 0.39 ^ab^	95.14 ± 0.33 ^a^	2.46 ± 0.25 ^b^
	6%	2.03 ± 0.04 ^b^	94.92 ± 0.45 ^a^	3.05 ± 0.44 ^b^
	8%	1.92 ± 0.28 ^b^	95.3 ± 0.56 ^a^	2.78 ± 0.31 ^b^
300 W	0%	96.59 ± 0.44 ^a^	2.31 ± 0.22 ^b^	1.1 ± 0.22 ^b^
	2%	95.51 ± 0.5 ^bc^	2.3 ± 0.32 ^b^	2.2 ± 0.28 ^a^
	4%	96.4 ± 0.08 ^ab^	2.22 ± 0.17 ^b^	1.38 ± 0.15 ^b^
	6%	94.58 ± 0.65 ^c^	3.7 ± 0.44 ^a^	1.72 ± 0.34 ^ab^
	8%	94.82 ± 0.64 ^c^	3.83 ± 0.41 ^a^	1.35 ± 0.32 ^b^
4%	300 W	96.4 ± 0.22 ^a^	2.2 ± 0.29 ^b^	1.4 ± 0.16 ^ab^
	500 W	96.65 ± 0.27 ^a^	2.36 ± 0.19 ^b^	0.99 ± 0.17 ^b^
	700 W	95.59 ± 0.35 ^b^	3.01 ± 0.14 ^a^	1.4 ± 0.32 ^a^
	900 W	94.92 ± 0.34 ^b^	3.28 ± 0.23 ^a^	1.8 ± 0.11 ^a^

Note: The different lowercase letter in the same row indicates that the means are significantly different at *p* < 0.05 according to the Duncan test.

**Table 2 foods-11-02066-t002:** Salt content of fresh shrimps after 8 h of curing with different salt concentrations and rehydration ratio of dried shrimps.

Salt Concentration (%)	NaCl Content (%)	MW Power (W)			
		300	500	700	900
0	0.54 ± 0.08 ^d^	1.72 ± 0.05 ^aB^	1.73 ± 0.06 ^aB^	1.86 ± 0.01 ^abA^	1.74 ± 0.06 ^aB^
2	0.83 ± 0.08 ^c^	1.73 ± 0.03 ^aA^	1.7 ± 0.15 ^aA^	1.72 ± 0.1 ^bA^	1.72 ± 0.04 ^aA^
4	1.24 ± 0.1 ^b^	1.71 ± 0.05 ^aAB^	1.65 ± 0.05 ^aB^	1.81 ± 0.06 ^abA^	1.74 ± 0.03 ^aAB^
6	1.32 ± 0.21 ^b^	1.78 ± 0.04 ^aB^	1.75 ± 0.02 ^aB^	1.9 ± 0.09 ^aA^	1.78 ± 0.04 ^aB^
8	1.83 ± 0.1 ^a^	1.77 ± 0.03 ^aB^	1.76 ± 0.01 ^aB^	1.87 ± 0.01 ^abA^	1.77 ± 0.04 ^aB^

Note: different lowercase letters “^a^”, “^b^”, “^c^”, and “^d^” in the same microwave power show that the index was significantly different (*p* < 0.05) at different salt concentration. Different uppercase letters “^A^” and “^B^” in the same salt concentration show that the index was significantly different (*p* < 0.05) at different microwave powers.

**Table 3 foods-11-02066-t003:** Effects of salt concentrations and MW powers on texture of dried shrimps after MW drying.

MW Power (W)	Salt Concentration (%)	Hardness/N	Springiness/mm	Chewiness/mj
300	0	13,073.57 ± 459.25 ^cB^	0.74 ± 0.03 ^bC^	6645.24 ± 742.37 ^bC^
	2	15,017.03 ± 352.93 ^bB^	0.77 ± 0.05 ^bA^	6596.82 ± 3291.31 ^bA^
	4	15,315.48 ± 115.71 ^bC^	0.76 ± 0.01 ^bA^	6181.45 ± 2484.28 ^bA^
	6	15,672.6 ± 495.17 ^bC^	0.75 ± 0.04 ^bB^	6226.52 ± 3461.97 ^abB^
	8	17,377.07 ± 165.73 ^aC^	0.9 ± 0.01 ^aA^	12,898.58 ± 439.85 ^aAB^
500	0	13,370.73 ± 382.3 ^cB^	0.82 ± 0.02 ^aAB^	7418.47 ± 435.81 ^abC^
	2	14,110.08 ± 357.07 ^cB^	0.79 ± 0.07 ^aA^	7273.6 ± 1565.97 ^abA^
	4	15,342.11 ± 737.51 ^bcC^	0.68 ± 0.1 ^abA^	5330.66 ± 3011.26 ^abA^
	6	16,608.35 ± 1615.88 ^abC^	0.54 ± 0.01 ^bC^	8749.91 ± 1221.77 ^bC^
	8	17,856.29 ± 511.98 ^aC^	0.83 ± 0.1 ^aA^	10,238.14 ± 3322.58 ^aB^
700	0	16,095.26 ± 736.28 ^dA^	0.87 ± 0.02 ^aA^	10,242.23 ± 1136.11 ^aA^
	2	18,042.5 ± 556.24 ^cA^	0.79 ± 0.06 ^aA^	9694.33 ± 133.31 ^aA^
	4	19,305.52 ± 434.79 ^cB^	0.79 ± 0.03 ^aA^	10,032.09 ± 93.04 ^aA^
	6	20,841 ± 859.17 ^bB^	0.74 ± 0.02 ^aAB^	10,179.21 ± 154.52 ^aB^
	8	23,141.2 ± 216.14 ^aB^	0.78 ± 0.09 ^aA^	12,255.06 ± 3364.41 ^aAB^
900	0	17,109.41 ± 1536.79 ^cA^	0.74 ± 0.06 ^bcC^	7951.27 ± 1236.1 ^bAB^
	2	18,965.85 ± 1298.36 ^cA^	0.72 ± 0.03 ^cA^	9196.86 ± 826.01 ^bA^
	4	21,919.57 ± 134.57 ^bA^	0.7 ± 0.08 ^cA^	9919.01 ± 2931.05 ^bA^
	6	24,556.5 ± 1560.21 ^abA^	0.85 ± 0.02 ^abA^	16,732.83 ± 2071.49 ^aA^
	8	24,932.99 ± 372.49 ^aA^	0.88 ± 0.01 ^aA^	18,802.32 ± 864.23 ^aA^

Note: different lowercase letters “^a^”, “^b^”, “^c^”, and “^d^” in the same microwave power show that the index was significantly different (*p* < 0.05) at different salt concentrations. Different uppercase letters “^A^”, “^B^” and “^C^” in the same salt concentration show that the index was significantly different (*p* < 0.05) at different microwave powers.

## Data Availability

Data is contained within the article.

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
