# Peer review of "Improvement of Pacific White Shrimp (Litopenaeus vannamei) Drying Characteristics and Quality Attributes by a Combination of Salting Pretreatment and Microwave"

_foods, 2022, doi:10.3390/foods11142066_

Round 1

Reviewer 1 Report

The paper is interesting. However, there are many syntax errors in the use of the English language, sentence construction, and grammatical errors, which need to be corrected. 

1. Page 2, lines 91-94. This sentence does not make sense.

2. Page 3, lines 120-122. This sentence also needs to be rewritten correctly.

3. Page 3, lines 123-130. Too long a sentence. Break it down into shorter and more meaningful sentences.

4. Page 4, lines 154-156. Please provide the PRC reference standard and a brief description of the method.

5. Page 5, lines 187-189. What is the software used for PCA? and give a reference.

6. Page 5, lines 192-194. Poor sentence construction. Rewrite succinctly to bring out the meaning.

7. Page 5, lines 220. The dried samples were observed by scanning electron microscopy (JSM-IT200, 220 JEOL, Japan).

8. Page 6, Figure 1. How did you get moisture readings before zero time of drying?

9. Page 6, lines 256-259. I cannot agree with this statement. At least it does not seem to be true by looking at Figure 1.

10. Pages 7/8, Figure 2. Could you indicate the bound, immobilized, and free water in the graphs?

11. Page 8, Table 1. This table is not clear. Does the first row refer to bound, immobilized, and free water in non-microwave dried samples? 

12. Page 9, Table 2. What are the salt concentration and salt content expressed in the table?

Also, the differences in the re-hydration ratios of shrimp cured in different salt concentrations are not obvious. They all look insignificant to me.

13. Page 9, lines 327-329. Certainly, this statement is not true as evident from Table 2. 

14. Page 10, lines 346-350. The error bars in these graphs are so high, that I do not think the values are statistically significant for you to make such a  conclusion. Please recheck the calculations.

Author Response

Dear editor Qi and reviewer 1,

Thank you for your consideration of our manuscript entitled “Salting Pretreatment Combined with Microwave Improves Drying Characteristics and Physicochemical Quality Attributes of Pacific White Shrimp (Litopenaeus VannameiRevealed from Water Distribution and Protein Secondary Structure”. We greatly appreciate you for the critical reading of our manuscript and giving us the instructive comments and suggestions.

We have carefully proof-read and revised the manuscript in accordance with the reviewers’ comments. The responses to the comments are described point-by-point as follows with the page and line numbers corresponding to the new or original manuscript. In particular, our answers were marked in red color in this response letter and the changes were marked in blue color in the revised manuscript.

Reviewer #1: General comments:

The paper is interesting. However, there are many syntax errors in the use of the English language, sentence construction, and grammatical errors, which need to be corrected.

Response: Thank you for your valuable comments. We have improved the use of English language, sentence construction, and grammatical errors.

Specific comments:

Point 1: Page 2, lines 91-94. This sentence does not make sense.

Response 1: We have revised the sentence. Please refer to Line 94-96 for details.

Point 2: Page 3, lines 120-122. This sentence also needs to be rewritten correctly.

Response 2: We have revised the sentence. Please refer to Line 120-121 for detail.

Point 3: Page 3, lines 123-130. Too long a sentence. Break it down into shorter and more meaningful sentences.

Response 3: We have shortened the sentence and expressed the critical meaning more clearly. Please refer to Line 122-129 for detail.

Point 4: Page 4, lines 154-156. Please provide the PRC reference standard and a brief description of the method.

Response 4: Thank you for your valuable comment. We have added relevant description. Please refer to Line 156-175 for detail.

Point 5: Page 5, lines 187-189. What is the software used for PCA? and give a reference.

Response 5: Thank you for your valuable comment. We have added relevant content. Please refer to Line 243-245 for detail.

Point 6: Page 5, lines 192-194. Poor sentence construction. Rewrite succinctly to bring out the meaning.

Response 6: We have revised the sentence. Please refer to Line 210-212 for detail.

Point 7: Page 5, lines 220. The dried samples were observed by scanning electron microscopy (JSM-IT200, 220 JEOL, Japan).

Response 7: Thank you for your valuable comment. We have revised the sentence. Please refer to Line 237-238 for detail.

Point 8: Page 6, Figure 1. How did you get moisture readings before zero time of drying?

Response 8: We have revised Figure 1. Please refer to Figure 1 for detail.

Point 9: Page 6, lines 256-259. I cannot agree with this statement. At least it does not seem to be true by looking at Figure 1.

Response 9: We have revised the sentences to make them more clearly. Please refer to Line 265-266 and 275-278 for details.

Point 10: Pages 7/8, Figure 2. Could you indicate the bound, immobilized, and free water in the graphs?

Response 10: Thank you for your valuable comment. We have added relevant information in Figure 2. Please refer to Figure 2 for detail.

Point 11: Page 8, Table 1. This table is not clear. Does the first row refer to bound, immobilized, and free water in non-microwave dried samples?

Response 11: Thank you for your valuable comment. The first row refers to to bound, immobilized, and free water in non-microwave dried sample. We have revised “Drying conditions” to “Treatment conditions”. Please refer to Table 1 for detail.

Point 12: Page 9, Table 2. What are the salt concentration and salt content expressed in the table? Also, the differences in the re-hydration ratios of shrimp cured in different salt concentrations are not obvious. They all look insignificant to me.

Response 12: Thank you for your valuable comment. The salt concentration refers to salt addingfor salting shrimp (0%,2%,4%,6% and 8%), and salt content refers to the content of NaCl contained in the shrimp after salting. We have changed “salt content” to “NaCl content”. Please refer to Table 2 for detail. The differences in the re-hydration ratios of shrimp cured in different salt concentrations are indeed not obvious. We have revised the sentence. Please refer to Line 345-346 for detail.

Point 13: Page 9, lines 327-329. Certainly, this statement is not true as evident from Table 2.

Response 13: We have revised the sentence to make it clearer. Please refer to Line 345-346 for details.

Point 14: Page 10, lines 346-350. The error bars in these graphs are so high, that I do not think the values are statistically significant for you to make such a conclusion. Please recheck the calculations.

Response 14: Thank you for your valuable suggestion. We have checked the calculations and reconstructed Figure 3.

Reviewer 2 Report

This is a complete research manuscript with valuable information.

On the basis of the presented studies, the beneficial effect of the pre-treatment of salting and microwave power (MW) on the possibility of shortening the drying time and different properties of white shrimps from the Pacific was demonstrated. A number of modern markings are included. Water distribution and textural features were assessed, including secondary protein structure, as well as volatile constituents, astaxanthin, and color parameters.

The presented issues are very valuable and should be used for further research and practical application. The manuscript is prepared properly and the issues are discussed extensively. The methodology is properly presented. Figures and tables are well prepared and cited. The results are well compiled and discussed, including statistical analysis. It was based on 47 literature items, nearly 40% of which are from recent years (2019-2022). It should be emphasized that the authors very synthetically presented issues that are closely related to the subject of the manuscript. 

The title of this manuscript could be changed, it is long and unusual.

Author Response

Reviewer #2: General comments:

This is a complete research manuscript with valuable information. On the basis of the presented studies, the beneficial effect of the pre-treatment of salting and microwave power (MW) on the possibility of shortening the drying time and different properties of white shrimps from the Pacific was demonstrated. A number of modern markings are included. Water distribution and textural features were assessed, including secondary protein structure, as well as volatile constituents, astaxanthin, and color parameters. The presented issues are very valuable and should be used for further research and practical application. The manuscript is prepared properly and the issues are discussed extensively. The methodology is properly presented. Figures and tables are well prepared and cited. The results are well compiled and discussed, including statistical analysis. It was based on 47 literature items, nearly 40% of which are from recent years (2019-2022). It should be emphasized that the authors very synthetically presented issues that are closely related to the subject of the manuscript. The title of this manuscript could be changed, it is long and unusual.

Response: Thank you for your valuable comments. We have revised the title. Please refer to Line 1-4 for detail.

Reviewer 3 Report

The submitted article studied dried Pacific white shrimps' drying characteristics and quality under different salt concentrations (0, 2%, 4%, 6%, 8%) and MW powers (300, 500, 700, 900 W). In addition, moisture distribution, microstructure, and protein secondary structure were analyzed to reveal the mechanism by which salt treatment affects drying characteristics and quality. The work is generally well written; however, in my opinion, several aspects require minor correction. Below are my comments on the individual sections of the manuscript:

Page 1, line 15, please specify shrimp species in Latin;

Page 2, line 45, please specify moisture content;

Page 3, line 124, were the samples mixed during the salt pretreatment?

Page 3, line 126, please specify the duration of MW drying for every oven power;

Page 5, lines 224-229, please add PCA to the Statistical Analysis section;

Page 10, consider adding a Color correlation diagram before PCA analysis of just provide correlation coefficients;

Page 11, consider merging diagram a and diagram b;

Page 11, line 383, please include variable contributions;

Page 13, please include result comparison with literature for astaxantin analysis;

Page 15, please provide a proposal for the future research in the conclusion section.

Author Response

Dear editor Qi and reviewer 3,

Thank you for your consideration of our manuscript entitled “Salting Pretreatment Combined with Microwave Improves Drying Characteristics and Physicochemical Quality Attributes of Pacific White Shrimp (Litopenaeus VannameiRevealed from Water Distribution and Protein Secondary Structure”. We greatly appreciate you for the critical reading of our manuscript and giving us the instructive comments and suggestions.

We have carefully proof-read and revised the manuscript in accordance with the reviewers’ comments. The responses to the comments are described point-by-point as follows with the page and line numbers corresponding to the new or original manuscript. In particular, our answers were marked in red color in this response letter and the changes were marked in blue color in the revised manuscript.

Reviewer #3: General comments:

The submitted article studied dried Pacific white shrimps' drying characteristics and quality under different salt concentrations (0, 2%, 4%, 6%, 8%) and MW powers (300, 500, 700, 900 W). In addition, moisture distribution, microstructure, and protein secondary structure were analyzed to reveal the mechanism by which salt treatment affects drying characteristics and quality. The work is generally well written; however, in my opinion, several aspects require minor correction.

Response: Thank you for your valuable comments. We have revised our manuscript and made improvements to the questions to better present the logical flow and explain the implications of our study. Grammar was also checked.

Specific comments:

Point 1: Page 1, line 15, please specify shrimp species in Latin;

Response 1: Thank you for your valuable comment. We have revised the sentence. Please refer to Line 15-16 for detail.

Point 2: Page 2, line 45, please specify moisture content;

Response 2: Thank you for your valuable comment. We have added relevant content. Please refer to Line 47-49 for detail.

Point 3: Page 3, line 124, were the samples mixed during the salt pretreatment?

Response 3: The samples were mixed every 2 h during the salt pretreatment. We have revised the sentence to provide more detailed information. Please refer to Line 122-124 for detail.

Point 4: Page 3, line 126, please specify the duration of MW drying for every oven power;

Response 4: Thank you for your valuable comment. We have added relevant content. Please refer to Line 126-129 for details.

Point 5: Page 5, lines 224-229, please add PCA to the Statistical Analysis section;

Response 5: Thank you for your valuable suggestion. We have added relevant information. Please refer to Line 243-245 for detail.

Point 6: Page 10, consider adding a Color correlation diagram before PCA analysis of just provide correlation coefficients;

Response 6: Thank you for your valuable comment. We added relevant content. Please refer to Figure 4 , Line 368-372 and 500-502 for detail.

Point 7: Page 11, consider merging diagram a and diagram b;

Response 7: Thank you for your patient reading of our manuscript. In this study, the drying quality of Pacific white shrimp were explored in terms of microwave power and salt concentration in this article as a whole, we have used two figures here in order to match the overall logic of this article.

Point 8: Page 11, line 383, please include variable contributions;

Response 8: Thank you for your careful reading. We have written the variable contributions in the text, and marked the part of the content that showed the variable contributions in blue. Please refer to Line 387-390 and 401-402 for details.

Point 9: Page 13, please include result comparison with literature for astaxantin analysis;

Response 9: Thank you for your valuable comment. We have provided more information and please refer to Line 453-459 for detail.

Point 10: Page 15, please provide a proposal for the future research in the conclusion section.

Response 10: Thank you for your great comment. We added relevant sentence and please refer to Line 525-528 for detail.

Round 2

Reviewer 1 Report

You have done most of the corrections that I suggested and the paper now reads better. However, there are still a few errors in the construction of sentences.